# The relationship between management practices and health facility performance: Evidence from low-resource, community-based facilities providing HIV services to key populations

Andrea Salas-Ortiz[1]*, Marjorie Opuni[2], José Luis Figueroa[3], Louis Masankha Banda[4], Alice Olawo[5], Spy Munthali[6], Julius Korir[7], Barbara Nyambura Thirikwa[8], Agatha Bula[9], Navindra Persaud[10], Sergio Bautista-Arredondo[3,11]

1 Centre for Health Economics, University of York, York, United Kingdom, 2 Independent Consultant, Geneva, Switzerland, 3 Division of Health Economics and Health Systems Innovations, National Institute of Public Health (INSP), Cuernavaca, Mexico, 4 FHI 360, Lilongwe, Malawi, 5 FHI 360, Nairobi, Kenya, 6 University of Malawi, Zomba, Malawi, 7 Kenyatta University, Nairobi, Kenya, 8 Independent Consultant, Nairobi, Kenya, 9 UNC Project, Lilongwe, Malawi, 10 FHI 360, Washington, DC, United States of America, 11 Economic Analysis Unit, Ministry of Health, Mexico City, Mexico

* andrea.salasortiz@york.ac.uk

## Abstract

Management practices are deemed especially critical to performance in resource-constrained settings. However, empirical evidence is limited for community-based organizations (CBOs) that provide health services. We studied the association between management domains and performance among organizations delivering HIV services to key populations. We assessed organizational performance using cost efficiency, service volume, and outreach service quality. We collected data on 67 management practices from 45 CBOs in Kenya and Malawi and classified them into six domains—community engagement, financial management, operations management, people management, performance monitoring, and target setting. We calculated the scores for each domain, adjusting them for contextual characteristics to control for observed differences across organizations. Using ordinary least squares and quantile regression models, we explored the relationship between management and performance. We also estimated the relative contribution of each management domain to the explained variation of all performance indicators via a Shapley decomposition approach. Better management was associated with lower unit costs for antiretroviral therapy (ART) ($-1,124 *pval*<0.05), HIV testing services (HTS) ($-45 *pval*<0.05), and sexually transmitted infection (STI) screening, ($-30 *pval*<0.01). Higher management scores correlated positively with HTS volume (2,806, *pval*<0.001) and outreach quality (258, *pval*<0.05). Financial management was the most important domain, explaining 13–49% of performance indicator variation, performance monitoring explained 31% of outreach quality variation, and target setting explained 23–34% of service volume variation. These findings suggest that targeted capacity building in

**Data availability statement:** The data underlying the results presented in the study are available from Harvard Dataverse (https://dataverse.harvard.edu/dataset.xhtml?persistentId=doi:10.7910/DVN/NUWFU9).

**Funding:** This study was funded by the Bill and Melinda Gates Foundation (OPP1175038). The funding agency had no role in the study design, collection/analysis of data, data interpretation, or manuscript writing.

**Competing interests:** The authors have declared that no competing interests exist.

management practices, especially financial management, performance monitoring, and target setting, could improve the provision of HIV health services to key populations by CBOs.

## Introduction

Financial resources for HIV programs have been declining in recent years. This decline has contributed to significant gaps between available resources and the funding required to achieve global targets [1]. The financial resource gap is especially large for programs aimed at key populations at higher risk of HIV infection in low-income countries. This includes programs for men who have sex with men, female sex workers, people who inject drugs, and transgender people, who have historically been under-resourced [1,2]. Community-based approaches implemented by community-based organizations (CBOs) are often employed to deliver effective HIV services to key populations [3,4], as social norms that discriminate against vulnerable groups can inhibit these populations from accessing HIV services [1]. While these approaches require additional funding to meet global objectives, optimizing facility performance at current resource levels is equally crucial.

Strong management is considered essential for the optimal performance of health facilities [5]. Management practices are deemed especially critical to organizational performance in resource-limited settings. This is because efficient use of scarce resources is vital, and strong management can facilitate "the achievement of large ends with limited means" [5]. However, information is urgently required on how management practices impact performance for policy decisions, including ways to improve performance by various stakeholders. There is a growing number of empirical studies on the association between management practices in health facilities and facility performance. Studies conducted in high-income countries have shown that better-managed hospitals (those with higher scores) are more productive [6,7] and have better quality of care [7–11]. Studies undertaken in low- and middle-income countries have found that better-managed primary health facilities have better quality of care. These better-managed facilities (those with higher scores) consistently outperform less-well-managed facilities [12–15]. In HIV service delivery, while one study in health facilities providing a mix of HIV services to the general population in low-resource countries found that some management practices were associated with better facility performance [16], another study focused on facilities providing male circumcision for HIV prevention found no statistically significant relationship [17]. This discrepancy may be due to the different outcomes analyzed, as well as the methodologies followed by each study.

None of the empirical research on the association between management practices and health facility performance conducted to date included facilities owned or operated by CBOs. In previous work, we analyzed the Linkages Across the Continuum of HIV Services for Key Populations Affected by HIV (LINKAGES) program in Kenya and Malawi. We found considerable variation in management practices among

facilities that provide HIV services to key populations, with some facilities showing very high management levels [18]. Within this sample, we also found significant differences in various measures of facility performance, including service costs and the volume of services provided [19,20]. Building on this foundation, in this paper, we study the associations between the management practices and the organizational performance of these CBOs delivering HIV services to key populations in Kenya and Malawi. These management practices are community engagement, financial management, operations management, people management, performance monitoring, and target setting. We assess organizational performance through measured proxies for cost efficiency, service volume, and outreach service quality. Specifically, we use the unit costs of antiretroviral therapy (ART), HIV testing services (HTS), and sexually transmitted infection (STI) screening as measures of cost efficiency; the number of key populations on ART and the numbers of HTS and STI screenings provided as measures of service volume [21,22]. The number of condoms provided per person reached through outreach services was used as an indicator of outreach service quality [23].

This paper contributes to the growing literature on the association between management practices and health facility performance in low-resource settings. It provides information on this relationship among unstudied, but paramount organizations for the provision of HIV health services for key populations—CBOs [24,25].

## Methods

We adopted a multi-country cross-sectional observational study design with three objectives: a) to identify variation of management practices across CBOs, b) to analyze the relationship between management and performance at the mean and across performance indicator distributions, and c) to identify which management domains matter the most for organizational performance.

### Context and data

Kenya and Malawi were purposively selected in consultation with program implementers. Both countries have generalized HIV epidemics with concentrated sub-epidemics among key populations [26]. In 2024, HIV prevalence in adults aged 15–49 years was estimated to be 3% in Kenya [27] and 6.2% in Malawi [28]. In Kenya, the latest HIV prevalence data for female sex workers and transgender people were 27.5% (2024) and 22% (2024) [27]. In Malawi, the latest HIV prevalence data for FSWs was 49.9% (2020) [28]. In both countries, key populations face important structural barriers that increase their risk of HIV infection and limit their access to health services [29].

From 2014 (in Malawi) and 2016 (in Kenya) to 2021, the LINKAGES program, funded by the United States Agency for International Development (USAID) through the United States President's Emergency Plan for AIDS Relief (PEPFAR) and administered by FHI 360, provided a comprehensive package of HIV services to key populations. This package was comprised of clinical services including post-exposure prophylaxis, pre-exposure prophylaxis, HIV testing services, antiretroviral therapy, sexually transmitted infection services, sexual and reproductive health services, and management of sexual violence. It also included non-clinical interventions—including empowerment and engagement services, structural interventions, and peer outreach—and pre-service activities—including population mapping and size estimation, and above-service management and monitoring. HIV services were delivered by local CBOs, called implementing partners (IPs), which were overseen by country offices and program headquarters in the United States (US). The IPs provided services in facilities called drop-in centers (DICs) and in communities.

The data analyzed in this paper come from a study conducted in a sample of 45 CBOs over two years. This sample reflected the multilevel LINKAGES program implementation structure, which consisted of a program headquarters; 2 country offices (1 per country); 18 IPs in Kenya and 2 in Malawi; and 30 DICs in Kenya and 15 in Malawi. This study builds on previous work done within a larger research project. In the first phase of the study, a costing study of the LINKAGES program in both countries was conducted for the US Government FYs 2018 and 2019 (1 October to 30 September). The methods and results of this work have been detailed elsewhere [19,20]. Results indicated significant variation in

organizational performance indicators (costs, service volume and quality) across DICs. We then aimed to explore whether these results might be due to differences in management practices across DICs. Thus, in the second phase, an online facility management survey targeting DIC managers was conducted from 1 November 2021–31 January 2022, with questions pertaining to management practices implemented in the DIC (or CBO) during FY 2019. The methods and descriptive results of the survey are documented elsewhere [18].

## Ethical clearance

The study received ethical approval from the ethical review board of the National Institute of Public Health of Mexico (Number: 1554), the Kenya Medical Research Institute and the National Commission for Science, Technology, and Innovation (Protocol No. 4258), and the National Commission on Research Ethics in the Social Sciences and Humanities of Malawi (Protocol No. P/07/21/590). All DIC managers who participated in the management survey completed an electronic informed consent form. Throughout the analysis, the authors did not have access to information that could identify individual participants.

## Management measure

Six management domains are explored, four—target setting, performance monitoring, people management, and operations management—based on the World Management Survey (WMS) framework [30,31]. These were expanded to include two additional domains that have been described as essential to quality management in resource-limited settings [32]—financial management [5] and community engagement [5,15].

As described previously [18], we collected data on 67 management practices classified into six domains—target setting (7 practices), performance monitoring (18 practices), people management (20 practices), operations management (11 practices), financial management (7 practices), and community engagement (4 practices). Scores for each dimension were calculated by aggregating each item depicted in S1 Table using Anderson's (2008) procedure. First, the item variables were normalized by demeaning and dividing each item by its standard deviation and obtaining the z-score. Second, weighted average scores across each domain were created using the inverse of the covariance matrix of the transformed outcome in each domain. For this approach, all items must follow a positive direction, consistently indicating a "better" outcome [33]. We calculated seven management scores for each CBO, including scores for the six dimensions and an aggregated management score (see S1 Table). The aggregated management score consolidates management capacity into a single indicator and reduces excessive statistical noise.

Next, we adjusted management practices for contextual characteristics to control for differences in scores across CBOs. Management practices depend on various factors [34], such as facility size [7], years of operational experience [35], location [36], and the level of educational training of those implementing the practices [7,37]. Competition or proximity to comparable sites also influences the implementation of management practices [38]. By controlling for these differences, management scores become comparable and isolated from these sources of variation, allowing us to focus on the relationship between performance and management. Adjusted management scores for each CBO were estimated as follows:

$$\hat{m}_i = \mathsf{E}\left[m_i | X_i\right] = X_i\beta + \in_i \tag{1}$$

Where $X$ represents the vector of contextual characteristics and $m_i$ the management score for dimension $m$ (1 = community engagement, 2 = financial management, 3 = operations management, 4 = people management, 5 = performance monitoring, and 6 = target setting) in DIC $i$, and $\hat{m}_i$ represents the adjusted score for each dimension. Equation (1) was estimated using ordinary least squares (OLS) regression models with robust standard errors. The same procedure was followed for the general management score, defined as $\hat{u}_i = E\left[u_i | X_i\right]$.

## Performance indicators

We analyzed three performance indicators relevant to CBOs that provide health services: unit cost, service volume, and outreach quality.

Unit cost represents the cost per health service provided and serves as a proxy for service delivery efficiency [39]. Unit costs for each intervention were calculated by summing up all input-related expenditures for providing an HIV service and dividing this by the number of services provided of that intervention. We previously calculated these costs, with details available in prior publications [19,20]. In short, we calculated economic costs by including all resources used to provide services, regardless of whether these incurred expenditures were for the LINKAGES program. To derive the total annual cost per DIC, we used weights to allocate above service delivery, pre-service delivery, and IP costs to DICs. For HTS, ART and STI screening, the unit cost per service was calculated by dividing the total costs of each intervention by the number of services provided for that intervention.

Service volume refers to the number of services provided each year, reflecting the productivity of the sites. Outreach quality is defined as the average number of condoms distributed per person reached. In the context of HIV services for key populations, outreach activities are paramount [3]. This indicator aims to capture the intensity of these activities and was previously proposed as a proxy for outreach quality along with the number of STI visits per person reached [23]. Due to data unavailability, in this study, we could only calculate the number of condoms distributed per person reached.

## Relationship between management and performance

Data on 67 management practices were collected from 45 CBOs (30 in Kenya and 15 in Malawi) in 2018 and 2019. However, management data are only available for FY 2019. We explored the relationship between adjusted management scores and performance indicators, assuming that management scores are unlikely to change significantly within one year and, thus, using repeated values of management scores within CBOs, and capitalizing mostly on between-variation.

We employed OLS regression models with robust standard errors, which we specified as follows:

$$Y_i = \alpha + \beta \hat{u}_i + \in_i \tag{2}$$

where $Y_i$ represents performance indicators ($Y$ = unit costs of HTS, ART, and STI screening services; the numbers of HIV tests, individuals on ART, and STI screenings; and the number of condoms per person reached) in facility $i$. We adjusted regression models of unit costs for the number of HIV services provided for each intervention, as it is common practice to control for variations in productivity when modelling unit costs [22,40,41]. For this initial exploratory exercise, we used the general adjusted management score ($\hat{u}$).

## Relative contribution of each management dimension to organizational performance

Most of the body of evidence about the association between management and organizational performance has found that monitoring, target setting and incentives are the most relevant practices [30,42]. However, these studies have not focused on organizations, such as CBOs, that have unique structural features [35]. Thus, we make use of the Shapley decomposition technique [43] to identify the relative contribution of each management domain towards organizational performance. This exercise can enlighten our understanding of the differential response of management practices across different settings.

The technique consists of estimating the relative contribution of each management dimension to the overall variation of each performance indicator. This method builds on Equation (2), which depicts a linear relationship between performance and management, but now includes the six management dimensions, as follows:

$$Y_i = \alpha + \sum_{m=1}^{6} \beta_m \hat{m}_i^m + \in_i \tag{3}$$

We exploit the linear and additive properties of Equation (3) to identify the relative contribution of each management dimension ($\hat{m}^m$) to each performance indicator, $y$. These estimated contributions are order-independent and linearly additive. However, these indicators only provide insights into relative contributions and should not be interpreted causally. The intuitive idea of this technique is found in the appendix (S1 File), and a complete description is presented elsewhere [44].

**Going beyond the mean: Heterogeneous impact of management across the performance distributions**

Equation (2) estimates the average relationship between performance and management based on the conditional mean function $E(Y|M)$. Results from this analysis might provide an incomplete view of this relationship [45]. This is because mean-based analysis does not allow us to explore which CBOs (those with low, medium, high, or very high performance) benefit the most from management practices. This insight is valuable for targeting purposes, such as identifying CBOs where management improvements can have the greatest impact. By estimating models at both the top and bottom of the performance indicator distributions, we gain a better understanding of the role of management practices across the entire distribution of indicators.

For this, we use quantile regression models (QRM), and denote the $q^{th}$ conditional quantile function of performance indicators, $Y$, given management, $\hat{M}$, as $Q_q\left(Y_i\middle|\hat{M}_i\right)$, and estimate its relationship as follows:

$$Q_q\left(Y_i\middle|\hat{M}_i\right) = \gamma_1 + \gamma_2\hat{M}_i + F^{-1}_{\in_i}(q)$$

(4)

where $F_{\in_i}$ denotes the distribution function of $\in_i$. We estimate Equation (4) for the $q$ = 25th, 50th, 75th, and 95th percentiles, and test whether the role of management is statistically the same across these percentiles. In the case of unit costs, lower percentiles depict higher performance (e.g., q25 represents very high performance, as unit costs are the lowest). Conversely, for service volume and outreach quality, lower percentiles indicate lower performance, e.g., $q^{25}$ ($q^{95}$) represents very low (high) productivity and quality.

## Results

The following subsections present the results from the above analyses.

### CBO characteristics

Table 1 provides an overview of the characteristics of LINKAGES CBOs in Kenya and Malawi. Most of them were in Kenya, and more than half of the managers had either a bachelor's or a postgraduate degree. On average, CBOs have been operating for five years and have an average of seven people on staff. There were, on average, two other CBOs providing HIV services within a 30-minute driving radius. The average CBO provided 1,215 HIV services over the 2018 and 2019 fiscal years. There was some variation among them in the number of HIV tests conducted, key populations on ART, STI screenings, and the average annual number of condoms distributed per key population reached. There was also considerable heterogeneity among CBOs in the unit costs for HTS, ART, and STI screening.

### Variation in overall management scores

Fig 1 shows a comparison between crude and adjusted overall management z-scores. Adjusted scores tend to show less variation. The Kolmogorov-Smirnov test, which tests for differences in the distribution of both variables, indicates that the distributions are significantly different, D stat = 1.28 (p-val < 0.01). However, there is considerable variation in the overall management z-scores across CBOs both before and after adjusting for contextual characteristics. Similar comparisons for all management dimensions—community engagement, financial management, people management, performance monitoring, operations management, and target setting—are shown in S1 Fig.

**Table 1. Description of key variables, LINKAGES CBOs in Kenya and Malawi.**

| | N | Mean | Median | Minimum | Maximum |
|---|---|---|---|---|---|
| *CBOs contextual variables* | | | | | |
| Proportion of CBOs located in Malawi | 45 | 0.33 | | 0 | 1 |
| Proportion of CBOs located in Kenya | 45 | 0.67 | | 0 | 1 |
| Proportion of CBO managers with bachelor or postgraduate degree | 45 | 0.53 | | 0 | 1 |
| Number of years from CBO opening | 45 | 5 | 3 | 1 | 16 |
| Number of staff working at the CBO | 45 | 7 | 6 | 2 | 21 |
| Number of CBOs providing HIV services within 30-minute driving radius | 45 | 2 | 2 | 0 | 5 |
| Number of HIV services provided | 90 | 1,215 | 473 | 16 | 19,647 |
| *Unit Costs [a]* | | | | | |
| ART | 89 | $847 | $531 | $259 | $5,785 |
| HTS | 90 | $68 | $44 | $10 | $332 |
| STI screening | 90 | $39 | $25 | $4.9 | $188 |
| *Service volume* | | | | | |
| Number of persons on ART | 89 | 112 | 89 | 2 | 346 |
| Number of HIV testing services | 90 | 1,888 | 1,586 | 76 | 9,253 |
| Number of STI screenings | 90 | 2,273 | 2,012 | 398 | 6,880 |
| *Outreach quality* | | | | | |
| Number of condoms per person reached | 90 | 275 | 242 | 11 | 1,996 |

[a]Expressed in US-2019.

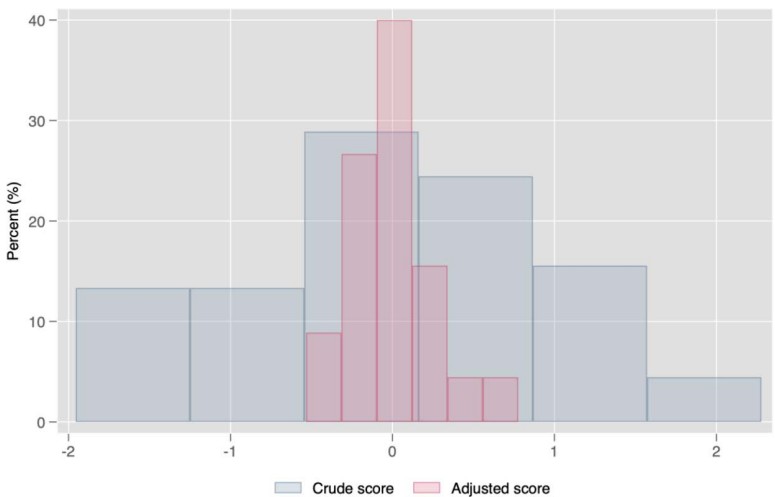

**Fig 1. Comparison of crude versus adjusted management z-scores across organizations.** Adjusted management practices for contextual characteristics to control for differences in scores across CBOs.

## Overall management scores and performance

Table 2 presents the results of the association between overall management scores and performance indicators—unit costs, service volume, and outreach quality. Lower management scores correlated negatively with the unit costs of providing ART, HTS, and STI screening. For a one standard deviation change in the management score, unit costs decreased

Table 2. Results from OLS model regressions between overall management and performance indicators.

| | Unit Costs[a] | | | Service volume | | | Outreach quality |
|---|---|---|---|---|---|---|---|
| Dependent variables | ART | HIV Testing | STI screening | Individuals on ART | HIV testing services | STI screenings | Condoms per person reached |
| Adjusted management score | −1,124.15* | −45.94* | −30.31** | 2.03 | 2,806.58*** | 992.62 | 258.03* |
| | (455.79) | (19.06) | (11.06) | (35.87) | (753.00) | (545.94) | (102.36) |
| Intercept | 1475.37*** | 98.64*** | 71.23*** | 111.51*** | 1888.37*** | 2273.04*** | 274.71*** |
| | (222.74) | (11.32) | (8.23) | (8.84) | (167.91) | (145.54) | (25.91) |
| Observations | 89 | 90 | 90 | 89 | 90 | 90 | 90 |
| Degrees of freedom | 7.94 | 13.02 | 15.17 | 0 | 13.89 | 3.31 | 6.35 |
| F-statistic p-val | <0.001 | <0.001 | <0.001 | 0.9 | <0.001 | 0.07 | 0.013 |

[a]Cost performance indicators adjusted for the scale of services provided in each intervention. Robust standard errors in parentheses. * $p < 0.05$, ** $p < 0.01$, *** $p < 0.001$.

$1,124 (pval<0.05), $45 (pval<0.05), and $30 (pval<0.01) for ART, HTS and STI, respectively. This indicates that better management is associated with lower unit costs for these services. The management score was positively associated with the number of individuals on ART, the number of HIV tests provided, and the number of STI screenings (2 pval>0.05, 2,806 pval<0.001, and 992 pval>0.05, respectively), suggesting that better management practices are associated with increases in service productivity (volume). However, this relationship was only statistically significant for the number of HIV tests provided. There was also a positive association between management score and the number of condoms distributed per person reached (258 pval<0.05), indicating that more management is related to higher levels of outreach quality.

## Contribution of management to performance

Table 3 shows the relative contributions of each management dimension—community engagement, financial management, people management, performance monitoring, operations management, and target setting—to the overall variation of each performance indicator. Financial management played a significant role across all indicators of CBO performance, particularly in explaining the variation in unit costs and the volume of HIV testing. Performance monitoring was the most important driver of outreach quality, while target setting explained most of the variation of individuals on ART and STI

Table 3. Relative contribution of each management dimension to the explained variation in different performance indicators.

| | Unit Costs[a] | | | Service volume | | | Outreach quality |
|---|---|---|---|---|---|---|---|
| | ART | HIV Testing | STI screening | Individuals on ART | HIV testing services | STI screenings | Condoms per person reached |
| Community engagement | 4.64 | 13.28 | 12.75 | 10.11 | 12.87 | 13.33 | 14.15 |
| Financial management | 13.09 | 30.78 | 26.14 | 17.02 | 48.56 | 26.29 | 20.69 |
| Operations management | 3.49 | 5.22 | 4.55 | 9.16 | 5.02 | 8.75 | 10.39 |
| People management | 6.73 | 5.29 | 2.52 | 7.68 | 6.81 | 6.97 | 9.98 |
| Performance monitoring | 10.50 | 8.25 | 8.10 | 21.89 | 4.08 | 14.07 | 30.79 |
| Target setting | 4.69 | 5.53 | 4.60 | 34.13 | 22.66 | 30.59 | 13.98 |

[a]For this indicator, the contribution of level of services scale is not shown. This does not alter the relative contributions of the management dimensions since Equation (3) displays additive separability and linearity. Magnitudes are shown in percentage terms (%).

screenings. Operations management and people management were the management dimensions that least explained the variation in all performance indicators.

**Relationship between management and performance beyond the mean**

The results of quantile linear regressions analyzing the relationships between management and performance at different parts of the performance indicator distributions are shown in Table 4. Unit costs and management were negatively associated, though most results were not statistically significant, indicating that for these indicators, management may not have different effects at different parts of the performance distribution.

For service volume and quality of services, the results were as follows. We found a strong and positive association between management and HIV testing service volume across all quantiles (25th = 1,564, *pval* < 0.001; 50th = 1,970, *pval* < 0.01; 75th = 3,393, *pval* < 0.01; 95th = 7,972, *pval* < 0.001). Our findings showed a positive association between management and STI screening volume, though only the 95th percentile was statistically significant (95th = 3,099, *pval* < 0.01). We also found statistically significant positive associations between management and outreach quality across the bottom three quantiles (25th = 189, *pval* < 0.05; 50th = 230, *pval* < 0.01; 75th = 343, *pval* < 0.001).

## Discussion

This study examined the relationship between management practices and organizational performance of community-based health facilities delivering HIV services to key populations as part of the LINKAGES program in Kenya and Malawi. Our findings suggest that more management practices, particularly those focused on financial management, performance monitoring, and target setting, are associated with better performance.

Studies on this relationship have found that the implementation and intensity of structured management practices are linked to higher levels of a firm's productivity, profitability, growth, survival rates, and innovation [46]. Better management practices were associated with improved cost efficiency for ART, HTS, and STI screening services, higher service productivity for HTS, and higher levels of outreach quality. Previous studies agree on the positive role of management practices on HIV unit costs [16,47], and process quality [16]. Lower unit cost suggests better resource use given a certain level of services provided, assuming quality remains constant [48]. Previous studies have shown that the existence of people and financial management practices correlated with higher levels of productivity and efficiency in community-based organizations [35,49]. However, to the best of our knowledge, this is the first paper that further investigates which performance indicator is driven the most by each of the different dimensions that constitute the spectrum of management practices. We found that the contribution of management to performance varied both by dimension and performance indicator. Financial management was a significant management dimension explaining substantial portions of the variation of all CBO performance indicators and was the most important dimension for the explained variation of unit costs and the volume of HIV testing. Performance monitoring was the most important driver of variation in outreach quality, while target setting explained most of the variation in individuals on ART and STI screenings. This coincides with results from previous studies showing that best-performing health sites are more likely to have higher management capacity, and potentially higher observed quality and efficiency in the provision of HIV health services [16,35,49].

The dominance of financial management across performance indicators may reflect the reality that CBOs operating with limited resources must optimize every funding stream to maintain service delivery. The importance of performance monitoring for outreach quality makes sense given that effective community engagement requires systematic tracking of contacts and service uptake. Target setting emerged as particularly important for service volume, likely because achieving clinical outcomes requires clear organizational goals. While the Shapley decomposition offers insights into these relative contributions, the results are indicative only and do not establish causal relationships between management dimensions and performance.

**Table 4. Results from the quantile linear regressions between management and performance.**

| Performance indicators | | Quantile of the performance indicator | Adjusted overall management score | SE | N | p-values across percentiles[a] | | |
|---|---|---|---|---|---|---|---|---|
| | | | | | | $q^{25}$ | $q^{50}$ | $q^{75}$ |
| **Unit Costs** | **ART** | q25 | −230.4** | −82.9 | 89 | | | |
| | | q50 | −234.3 | −158.8 | | 0. 97 | | |
| | | q75 | −292.9 | −498.8 | | 0.9 | 0.9 | |
| | | q95 | −3,250.8* | −1362.8 | | 0. 11 | 0. 11 | 0. 03 |
| | **HIV Testing** | q25 | −15.4 | −12.1 | 90 | | | |
| | | q50 | −31.6** | −10.5 | | 0.26 | | |
| | | q75 | −26.3 | −33.8 | | 0. 75 | 0. 86 | |
| | | q95 | −199.0 | −142.4 | | 0. 35 | 0. 40 | 0. 17 |
| | **STI screening** | q25 | −15.4 | −8.4 | 90 | | | |
| | | q50 | −17.3 | −8.8 | | 0. 79 | | |
| | | q75 | −19.3 | −10.7 | | 0. 76 | 0. 86 | |
| | | q95 | −93.6 | −60.0 | | 0. 45 | 0. 46 | 0. 21 |
| **Service volume** | **Persons on ART** | q25 | −31.6 | −31.6 | 89 | | | |
| | | q50 | −30.7 | −41.1 | | 0.98 | | |
| | | q75 | −7.5 | −70.4 | | 0.73 | 0.7 | |
| | | q95 | 128.6 | −114.0 | | 0.3 | 0.32 | 0.18 |
| | **HIV testing services** | q25 | 1,563.5*** | −415.5 | 90 | | | |
| | | q50 | 1,970.0** | −585.9 | | | | |
| | | q75 | 3,392.6** | −1236.6 | | 0. 56 | | |
| | | q95 | 7,972.3*** | −1592.4 | | 0. 16 | 0. 20 | |
| | | | | | | 0.00 | 0. 00 | 0. 00 |
| | **STI screenings** | q25 | 939.1 | −493.1 | 90 | | | |
| | | q50 | 185.9 | −668.2 | | 0.17 | | |
| | | q75 | 522.1 | −1091.3 | | 0.68 | 0.72 | |
| | | q95 | 3,099.1** | −1105.8 | | 0.12 | 0.06 | 0.05 |
| **Outreach quality** | **Condoms per person reached** | q25 | 189.0* | −80.3 | 90 | | | |
| | | q50 | 229.6** | −85.7 | | 0.65 | | |
| | | q75 | 343.2*** | −96.6 | | 0.17 | 0.23 | |
| | | q95 | 1,115.5 | −715.2 | | 0.23 | 0.26 | 0.25 |

[a]P-values testing the relationship between management and the outcome variables across different percentiles and obtained using bootstrap with 500 repetitions. SE means standard error. * $p < 0.05$, ** $p < 0.01$, *** $p < 0.001$.

The quantile regression analysis provides insights into how management effects vary across performance distributions. For HIV testing services, management practices showed consistent positive associations across all performance levels. This pattern suggests that high-performing CBOs may be better positioned to leverage management improvements for substantial gains in service delivery. In contrast, for outreach quality, management effects were significant only through the 75th percentile, indicating potential diminishing returns at the highest performance levels. These findings align with organizational theory, suggesting that management interventions may have differential impacts depending on baseline organizational capacity [36]. The limited statistically significant effects of management on unit costs across quantiles were notable, given that the overall regression analysis showed significant negative associations between management and unit costs for all three services. The lack of significant effects in the quantile analysis may be due to smaller sample sizes within each quantile, reducing statistical power to detect effects.

Several limitations should be kept in mind when interpreting our findings. The cross-sectional design with a small sample size and only one year of management data means we assume no changes in management over time, and findings cannot be interpreted in causal terms since we do not control for potential sources of endogeneity. Management practices were self-reported, and the two-year gap between collecting management and performance data may have introduced some bias. Our performance indicators do not capture all aspects of organizational effectiveness. The Shapley decomposition, though useful for understanding relative contributions, is descriptive and does not establish causal relationships. Finally, our sample of CBOs participating in the LINKAGES program may not fully represent all CBOs serving key populations

Despite these limitations, results from this analysis provide useful evidence on the relationship between management practices and performance in community-based HIV service delivery. The findings suggest that financial management, performance monitoring, and target setting may be particularly important areas for organizational development. These results could inform discussions about how to strengthen management capacity in community-based facilities serving key populations, though further research is needed to understand the mechanisms through which management practices influence performance and to evaluate targeted interventions.

Future research should explore the potential simultaneity in the relationship between management practices and performance and investigate the mechanisms through which management practices influence organizational performance. This could include examining the role of leadership capabilities and organization learning processes in translating management practices into improved outcomes [50,51]. Understanding the specific elements within financial management, performance monitoring, and target setting that drive performance improvements could also provide insights into optimizing resource use.

## Conclusion

This study demonstrates that management practices are associated with organizational performance among CBOs delivering HIV services to key populations in Kenya and Malawi. Financial management, performance monitoring, and target setting emerged as the most important domains for CBO performance. With increasing funding constraints, these results suggest where limited capacity-building resources might be most effectively directed. The evidence suggests that targeted capacity building in these three management domains could improve the provision of HIV health services to key populations by CBOs. Understanding the most effective approaches for building these management capacities and evaluating targeted interventions remains an important area for future research.

## Supporting information

**S1 Table. Items included in each management domain.** [a]As defined in [18]. All items coded as 1 = Yes, 0 = No.
(PDF)

**S1 File. Relative contribution of each management dimension to organizational performance.**
(PDF)

**S1 Fig. Comparison of crude versus adjusted scores for all management dimensions.** Adjusted management practices for contextual characteristics to control for differences in scores across CBOs.
(TIF)

## Acknowledgments

We are grateful for the support and helpful comments on this analysis received from colleagues from FIH360, and managers and staff of the LINKAGES program in Kenya and Malawi. We thank Alejandra Rodríguez-Atristain, Jorge Eduardo Sánchez-Morales and David Contreras-Loya for their assistance with data curation and analysis. We also thank the United States Agency for International Development (USAID) for supporting the LINKAGES program.

## Author contributions

**Conceptualization:** Andrea Salas-Ortiz, Sergio Bautista-Arredondo.

**Formal analysis:** Andrea Salas-Ortiz.

**Funding acquisition:** Sergio Bautista-Arredondo.

**Investigation:** Andrea Salas-Ortiz.

**Methodology:** Andrea Salas-Ortiz.

**Writing – original draft:** Andrea Salas-Ortiz, Marjorie Opuni.

**Writing – review & editing:** Andrea Salas-Ortiz, Marjorie Opuni, José Luis Figueroa, Louis Masankha Banda, Alice Olawo, Spy Munthali, Julius Korir, Barbara Nyambura Thirikwa, Agatha Bula, Navindra Persaud, Sergio Bautista-Arredondo.

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
