## [Decision Letter · Decision Letter 0]

9 Jul 2025

Dear Dr. Salas Ortiz,

Thank you for submitting your manuscript to PLOS ONE. After careful consideration, we feel that it has merit but does not fully meet PLOS ONE’s publication criteria as it currently stands. Therefore, we invite you to submit a revised version of the manuscript that addresses the points raised during the review process.

**ACADEMIC EDITOR: **

Include key statistical findings in the abstract.

We look forward to receiving your revised manuscript.

Kind regards,

Daniel Chukwuemeka Ogbuabor, Ph.D., M.D.

Academic Editor

PLOS ONE

Journal Requirements:

2. Please note that your Data Availability Statement is currently missing the repository name OR a direct link to access each database. If your manuscript is accepted for publication, you will be asked to provide these details on a very short timeline. We therefore suggest that you provide this information now, though we will not hold up the peer review process if you are unable.

Additional Editor Comments:

The authors should include key statistical findings in the abstract.

Reviewers' comments:

Reviewer's Responses to Questions

**Comments to the Author**

1. Is the manuscript technically sound, and do the data support the conclusions?

Reviewer #1: Yes

Reviewer #2: Yes

2. Has the statistical analysis been performed appropriately and rigorously?

Reviewer #1: Yes

Reviewer #2: Yes

3. Have the authors made all data underlying the findings in their manuscript fully available?

Reviewer #1: Yes

Reviewer #2: Yes

4. Is the manuscript presented in an intelligible fashion and written in standard English?

Reviewer #1: Yes

Reviewer #2: Yes

Reviewer #1: 1. General comment: I found the work highly insightful for publication; however, I have suggested some minor recommendations to make the work stronger.

2. Sectional comments

a. Abstract—This is well written, but there is a need to revise the conclusion to provide a brief summary of the key findings, potential implications, and the way forward.

b. Background: This text is also well composed; however, an additional paragraph following line 85 on page 4, which justifies the selection of the two countries supported by statistics aligned with the study's objectives, would be great.

c. Methods—This is ok, but the authors may consider adding an introduction of study design between lines 99 and 100. Despite the citation provided to the used technique elsewhere, a brief description of the sampling and selection criteria used to choose the DIC is necessary under line 117 on page 6. Also provide a rationale linking the current study to phases 1 and 2. On page 10, lines 204-205, the authors may anchor the application of the Shapley decomposition approach with a theory.

d. Results—the statistics used are sound, but values are not provided in the text. The authors need to include the correlation coefficient, DF, and p-value appropriately in variables in lines 261-274 (statistics values in the text), in lines 279-287, and in lines 295-307. Additionally, the CHO characteristics may be enriched with country-level analysis that may reveal in-depth dimensions essential to policymakers in each country. Figure 1 appears to be missing in line 255.

e. Discussion—This is well written, but the authors may provide a proposition to explain their finding in line 317. Also state the limitation for application of Shapley decomposition.

f. Conclusion: revise the conclusion to provide key finding, potential implications, and a scalable management way forward

General rubrics

• The authors may consider splitting long sentences into in line 49-51, 51-54, 59-61, 76-79, 82-85, 217-221

• Delete the word "of" in 225.

Reviewer #2: The abstract mentions that data were collected on "67 management practices..." while in the methods sections again reads "data were collected from 45 CBOs" It is unclear how these figures relate. Does "67" refere to individual management indicators or items measured within each CBO that is, 67 variables per CBO? If so, this should be explicitly stated to avoid confusion with the number of CBOs surveyed.

Below are observations per specific lines

Line 21: reads "corresponding autor" should be written " corresponding author"

Line 27-28: reads "we proxy organizational performance..." I suggest to read " we used proxy indicators for organizational performance..."

Line 34: reads "we thoroughly explored the relationship" i suggest replacing "thorough" with a more precise descriptor like statistically analyzed or just remove "thoroughly to read "we explored the relationship"

Line 68: reads "...were better managed.." could yiou briefly explain how "better managed" facilities were defined in those studies.

Line 69-73: reads "One study found association...another did not..." is it possible to mention why these discrepancies may have occurred"

Line 74-86: justifies focus on CBOs and introduces the LINKAGES program: could you also explain as t owhy CBOs are particularly relevant to KP service delivery, perhaps due to trust or accessibility? In addition, the rationale for focusing specifically on CBOs could be stated earlier. I suggest to move the justification earlier in the introduction for stronger framing

Line 187: reads "...2018 and 2029" it should read "2018 and 2019"

Line 250: reads "Although adjusted scores tend to show less variation..." is it possible to quantify "less" how significant is the difference?

Line 343: reads "...managers take action to ensure that issues are addressed" I suggest specifying types of actions taken e.g task shifting etc

Line 351: reads "At the margin of these limitations..." I suggest to read " Despite these limitations..."

**Do you want your identity to be public for this peer review?** For information about this choice, including consent withdrawal, please see our Privacy Policy

Reviewer #1: No

Reviewer #2: **Yes: ** Manasseh Joel Mwanswila

---

## [Author Response · Author response to Decision Letter 1]

17 Jul 2025

PLOS ONE

PONE-D-25-28030

17 July 2025

Dear Daniel Chukwuemeka Ogbuabor, Ph.D., M.D.

Academic Editor

“The relationship between management practices and health facility performance: evidence from low-resource, community-based facilities providing HIV services to key populations”.

We appreciate the time and effort that you and the reviewers spent reviewing our initial manuscript. The feedback has allowed us to refine our paper's content and presentation. We have revised our manuscript to address all the reviewers’ comments, and your comment regarding including key statistical findings in the abstract.

We have ensured that our manuscript meets PLOS ONE's style requirements and that the Data Availability Statement is reported. We have reviewed the reference list to ensure that it is complete and correct.

The submitted manuscript highlights all changes to the original text. We also include our point-by-point responses to the reviewers’ comments.

We thank you again for your time and consideration.

Yours sincerely,

Dr. Andrea Salas-Ortiz

Corresponding Author

Responses to Reviewers

We thank the Reviewers for their constructive comments and include below our responses in italics.

Reviewer: 1

General comment: I found the work highly insightful for publication; however, I have suggested some minor recommendations to make the work stronger.

1. Sectional comments

1.a. Abstract—This is well written, but there is a need to revise the conclusion to provide a brief summary of the key findings, potential implications, and the way forward.

Response: Thank you for the suggestion. We have edited the results and conclusions section of the abstract accordingly as follows:“Better management was associated with lower unit costs for antiretroviral therapy (ART) ($-1,124 pval<0.05), HIV testing services (HTS) ($-45 pval<0.05), and sexually transmitted infection (STI) screening, ($-30 pval<0.01). Management scores were positively correlated with HTS volume (2,806, pval<0.001) and outreach quality (258, pval<0.05). Financial management was the most important domain, explaining 13-49% of performance indicator variation, performance monitoring explained 31% of outreach quality variation, and target setting explained 23-34% of service volume variation. These findings suggest that targeted capacity building in management practices especially financial management, performance monitoring, and target setting could improve the provision of HIV health services to key populations by CBOs.”

1.b. Background: This text is also well composed; however, an additional paragraph following line 85 on page 4, which justifies the selection of the two countries supported by statistics aligned with the study's objectives, would be great.

Response: Kenya and Malawi were purposively selected for this study. We have specified this in the methods and added additional context for both countries: “Kenya and Malawi were purposively selected in consultation with program implementers. Both countries have generalized HIV epidemics with concentrated sub-epidemics among key populations [26]. In 2024, HIV prevalence in adults aged 15–49 years was estimated to be 3% in Kenya [27] and 6.2% in Malawi [28]. In Kenya, the latest HIV prevalence data for female sex workers and transgender people were 27.5% (2024) and 22% (2024) [27]. In Malawi, the latest HIV prevalence data for FSWs was 49.9% (2020) [28]. In both countries, key populations face important structural barriers that increase their risk of HIV infection and limit their access to health services [29].”

1.c. Methods—This is ok, but the authors may consider adding an introduction of study design between lines 99 and 100. Despite the citation provided to the used technique elsewhere, a brief description of the sampling and selection criteria used to choose the DIC is necessary under line 117 on page 6. Also provide a rationale linking the current study to phases 1 and 2. On page 10, lines 204-205, the authors may anchor the application of the Shapley decomposition approach with a theory.

Response: We appreciate these insightful comments. We have added information about the study design: “We adopted a multi-country cross-sectional observational study design with three objectives: a) to identify variation of management practices across CBOs, b) to analyze the relationship between management and performance at the mean and across performance indicator distributions, and c) to identify which management domains matter the most for organizational performance.”

As well as more details about the sampling process: “The data analyzed in this paper come from a study conducted in a sample of 45 CBOs over two years. This sample reflected the multilevel LINKAGES program implementation structure, which consisted of a program headquarters; 2 country offices (1 per country); 18 IPs in Kenya and 2 in Malawi; and 30 DICs in Kenya and 15 in Malawi. This study builds on previous work done within a larger research project.”.

As requested, by also described how this study builds on previous the previous phases of the research project, and also justified the application of the Shapley decomposition technique.

1.d. Results—the statistics used are sound, but values are not provided in the text. The authors need to include the correlation coefficient, DF, and p-value appropriately in variables in lines 261-274 (statistics values in the text), in lines 279-287, and in lines 295-307.

Additionally, the CHO characteristics may be enriched with country-level analysis that may reveal in-depth dimensions essential to policymakers in each country. Figure 1 appears to be missing in line 255.

Response: Thanks for pointing this out. We have added the missing information where indicated. Figure 1 has been submitted separately, as required by the journal.

We also agree with the reviewer about country-level analyses enriching the evidence for policymakers in Kenya and Malawi. However, these parametric analyses are unfeasible due to sample size issues.

1.e. Discussion—This is well written, but the authors may provide a proposition to explain their finding in line 317. Also state the limitation for application of Shapley decomposition.

Response: We have added the following paragraph to the discussion to address this concern. “The dominance of financial management across performance indicators may reflect the reality that CBOs operating with limited resources must optimize every funding stream to maintain service delivery. The importance of performance monitoring for outreach quality makes sense given that effective community engagement requires systematic tracking of contacts and service uptake. Target setting emerged as particularly important for service volume, likely because achieving clinical outcomes requires clear organizational goals. While the Shapley decomposition offers insights into these relative contributions, the results are indicative only and do not establish causal relationships between management dimensions and performance.”

1.f. Conclusion: revise the conclusion to provide key finding, potential implications, and a scalable management way forward

Response: We have rewritten the conclusion as suggested: “This study demonstrates that management practices are associated with organizational performance among CBOs delivering HIV services to KPs. Financial management, performance monitoring, and target setting emerged as the most important domains for CBO performance in Kenya and Malawi. With increasing funding constraints, these results suggest where limited capacity-building resources might be most effectively directed. The evidence suggests that targeted capacity building in these three management domains could improve the provision of HIV health services to KPs by CBOs. Understanding the most effective approaches for building these management capacities and evaluating targeted interventions remains an important area for future research.”

2. General rubrics

2.a The authors may consider splitting long sentences into in line 49-51, 51-54, 59-61, 76-79, 82-85, 217-221

Response: We have implemented these suggestions.

2.b Delete the word "of" in 225.

Response: Thanks for pointing this out. We have changed this.

Reviewer: 2

1. The abstract mentions that data were collected on "67 management practices..." while in the methods sections again reads "data were collected from 45 CBOs" It is unclear how these figures relate. Does "67" refere to individual management indicators or items measured within each CBO that is, 67 variables per CBO? If so, this should be explicitly stated to avoid confusion with the number of CBOs surveyed.

Response: We collected data on 67 management practices in 45 CBOs. We have clarified this to avoid any confusion.

2. Below are observations per specific lines

2.a Line 21: reads "corresponding autor" should be written " corresponding author"

Response: Thanks for pointing this out. We have changed this.

2.b Line 27-28: reads "we proxy organizational performance..." I suggest to read " we used proxy indicators for organizational performance..."

Response: We have implemented this suggestion.

2.c Line 34: reads "we thoroughly explored the relationship" i suggest replacing "thorough" with a more precise descriptor like statistically analyzed or just remove "thoroughly to read "we explored the relationship"

Response: We have implemented this suggestion.

2.d Line 68: reads "...were better managed.." could yiou briefly explain how "better managed" facilities were defined in those studies.

Response: Thanks for pointing this out. We now explain that those facilities with higher management scores were classified as better managed.

2.e Line 69-73: reads "One study found association...another did not..." is it possible to mention why these discrepancies may have occurred"

Response: This is an insightful comment. We have now explained that the discrepancy may be due to the different outcomes analysed, as well as the methodologies followed by each study.

2.f Line 74-86: justifies focus on CBOs and introduces the LINKAGES program: could you also explain as t owhy CBOs are particularly relevant to KP service delivery, perhaps due to trust or accessibility? In addition, the rationale for focusing specifically on CBOs could be stated earlier. I suggest to move the justification earlier in the introduction for stronger framing.

Response: We moved the justification earlier in the introduction and explained that CBOs have historically played an essential role in targeting key populations, as social norms that discriminate against vulnerable groups can inhibit these populations from accessing HIV and prevention services.

2.g Line 187: reads "...2018 and 2029" it should read "2018 and 2019"

Response: Thanks for pointing this out. We have changed this.

2.h Line 250: reads "Although adjusted scores tend to show less variation..." is it possible to quantify "less" how significant is the difference?

Response: Thanks for pointing this out. Given the units of the variables (z-scores), we applied the Kolmogorov-Smirnov test, which tests for differences in the distribution of both variables. The test indicates that both distributions are statistically significantly different. We added this to the manuscript.

2.i Line 343: reads "...managers take action to ensure that issues are addressed" I suggest specifying types of actions taken e.g task shifting etc

Response: Thanks for pointing this out. We have changed this.

2.j Line 351: reads "At the margin of these limitations..." I suggest to read " Despite these limitations..."

We have implemented this suggestion.

---

## [Editor Report · Decision Letter 1]

30 Jul 2025

The relationship between management practices and health facility performance: evidence from low-resource, community-based facilities providing HIV services to key populations

PONE-D-25-28030R1

Dear Dr. Andrea Salas Ortiz, 

We’re pleased to inform you that your manuscript has been judged scientifically suitable for publication and will be formally accepted for publication once it meets all outstanding technical requirements.

Kind regards,

Daniel Chukwuemeka Ogbuabor, Ph.D., M.D.

Academic Editor

PLOS ONE
---

## [Editor Report · Acceptance letter]

PONE-D-25-28030R1

PLOS ONE

Dear Dr. Salas Ortiz,

I'm pleased to inform you that your manuscript has been deemed suitable for publication in PLOS ONE. Congratulations! Your manuscript is now being handed over to our production team.

Kind regards,

on behalf of

Dr. Daniel Chukwuemeka Ogbuabor

Academic Editor

PLOS ONE